# CaCO_3_–Chitosan Composites Granules for Instant Hemostasis and Wound Healing

**DOI:** 10.3390/ma14123350

**Published:** 2021-06-17

**Authors:** Wei He, Xiaodong Huang, Jun Zhang, Yue Zhu, Yajun Liu, Bo Liu, Qilong Wang, Xiaonan Huang, Da He

**Affiliations:** 1Department of Spine Surgery, Beijing JiShuiTan Hospital, 4th Medical College of Peking University, Xicheng District, Xinjiekou No. 31 East Street, Beijing 100035, China; 18811060224@163.com (W.H.); drliuyajun@163.com (Y.L.); drliubo@163.com (B.L.); qlw2013@pku.edu.cn (Q.W.); 2Department of Orthopedics, The Third Affiliated Hospital of Guangzhou Medical University, 63 Duobao Road, Liwan District, Guangzhou 510150, China; huangxiaodong08@126.com; 3Department of Spine Surgery, Zhejiang Provincial People’s Hospital, Hangzhou Medical College People’s Hospital, Hangzhou 310014, China; spinezhangjun@aliyun.com; 4Department of Chemistry, Capital Normal University, Haidian District, Beijing 100035, China; 2170702071@cnu.edu.cn

**Keywords:** CaCO_3_–chitosan, hemostasis, wound healing

## Abstract

Excessive bleeding induces a high risk of death and is a leading cause of deaths that result from traffic accidents and military conflict. In this paper, we developed a novel porous chitosan–CaCO_3_ (CS–CaCO_3_) composite material and investigated its hemostatic properties and wound healing performance. The CS–CaCO_3_ composites material was prepared via a wet-granulation method. Granulation increases the infiltrating ability of the CS–CaCO_3_ composites material. The improved water absorption ability was enhanced to 460% for the CS–CaCO_3_ composites material compared to the CaCO_3_ or chitosan with only one single component. The coagulation studies in vivo illustrated that the blood clotting time was greatly reduced from 31 s for CaCO_3_ to 16 s for the CS–CaCO_3_ composite material. According to the results of the wound healing experiments in rats, it was found that the CS–CaCO3 composite material can promote wound healing. The CS–CaCO_3_ composite material could accelerate wound healing to a rate of 9 days, compared with 12 days for the CaCO_3_. The hemostatic activity, biocompatibility, and low cost of CS–CaCO_3_ composite material make it a potential agent for effective hemostatic and wound healing materials.

## 1. Introduction

Hemorrhaging is a leading cause of deaths that result from traffic accidents and military conflict [1,2]. In hospitals, hemorrhage causes 15–25% of trauma deaths. A lot of hemostatic materials are used for hemorrhage controlling [3,4], which can be divided into inorganic materials [5,6,7] and organic polymer materials [8]. Inorganic materials include WoundStat, QuikClot, and Combat Gauze and polymer materials include Celox, Celox-D, HemCon, which are all commercially available products for hemostatic wound healing. However, the inorganic materials have many disadvantages, e.g., in the first generation of inorganic hemostatic materials, QuikClot caused high exothermic reactions and induced tissue empyrosis and other abnormal foreign-body reactions [9,10], and another inorganic material, WoundStat, induced thrombus in lungs and vessels [11,12]. Additionally, the poor adhesion of Combat Gauze made more blood loss [13]. Therefore, research is necessary to find more effective hemostatic agents.

Thrombin is an enzyme which can support earlier fibrin generation in the “waterfall reaction” of blood clotting. Calcium ions have the function to form thrombin and then activate the coagulation system. At the same time, intracellular Ca^2+^ can cause phosphatidyl serines to be exposed on platelet surfaces, and further help to assemble coagulation complexes and amplify the formation of thrombin [14,15]. Ca^2+^ can also stop bleeding by increasing clot rigidity. Similar to the benefit of Ca^2+^, CaCO_3_ also has good biocompatibility and potential for applications in medicine. In traditional Chinese medicine, cuttlebone, a kind of traditional hemostatic material that consists of CaCO_3_ is orally administered and can rapidly stop bleed [16,17,18,19,20,21]. Accordingly, developed hemostatic materials based on CaCO_3_ have been investigated for tissue repairing and rapid wound closure [22,23]

Chitosan is an inexpensive natural-origin polymer that has been widely used in clinics for its biodegradability, biocompatibility and antimicrobial activities [24,25]. Chitosan also has been widely investigated because of its hemostatic function [26,27,28,29,30]. The protonated amine groups of chitosan have a static interaction with the negative charge of red blood cells and platelets that induces formation of thrombus [31]. Chitosan also has good water absorption ability to facilitate hemostasis by blood coagulation factors and concentrating platelets [32,33,34]. Some commercially available materials have been developed based on chitosan as hemostatic agents such as Celox, TraumaStat, HemCon and so on. Chitosan-based materials demonstrated effective and safe properties for trauma application [35]. However, chitosan hemostatic materials cannot be used to treat wounds with heavy bleeding, because chitosan materials have no adhesion and may be washed away by blood, or chitosan materials may enter the blood vessels of the wound and cause thrombosis [36].

In this paper, we developed a novel CaCO_3_ and chitosan composite material used for hemorrhage control and wound healing due to their hemostatic properties and biocompatibility [37,38]. CaCO_3_ was mixed via wet-granulation method on chitosan by precipitation to improve the hemostatic activity of the composite material. The novel CS–CaCO_3_ composite material was characterized for its physicochemical properties. Due to the positive charge of the protonated amine group of chitosan, it can form a plug with the negative charge of erythrocytes cell membranes induced by electrostatic interaction, and then activate a coagulation cascade that brings a rapid and effective hemostasis (Scheme 1). In this manner, the CS–CaCO_3_ composite material exhibited excellent hemostatic and wound healing property on the injured site.

## 2. Experimental

### 2.1. Chemicals

Na_2_CO_3_ and CaCl_2_ were purchased from Sinopharm Chemical Reagent Ltd. (Beijing, China) and used to synthesis CaCO_3_ xerogel. Chitosan (Mw: 100,000–300,000 Da, from shrimp shells, deacetylated ratio ≥ 75%), were purchased from Integrated Scientific and Industrial Resource Platform (Beijing, China) and used for materials preparation and coatings. The M_W_ of Chitosan was determined by the capillary viscometry method using an Ubbelohde viscometer Synthware Ltd. (Mainz, Germany)in the solution of 0.1 mol/L CH_3_COOH/0.2 mol/L NaCl at 25 °C, the viscosity average molecular weight (M_v_) was calculated with vale of 367 kDa using the Mark−Houwink equation: [η] = K_m_M_v_^α^, where K_m_ = 1.81 × 10^−3^, α = 0.93. The degree of deacetylation of CS was further determined by elemental analyses analysis on a Vario EL III CHN analyzer (Elementar Ltd. Stockport, UK) with the value of 77.8%. Fetal bovine serum was purchased from Invitrogen (Carlsbad, CA, USA). Other commercially available chemicals were used as received. High purity deionized water (18 MΩ) was used throughout this work, and all other chemicals used were of analytical grade.

### 2.2. Preparation of CS–CaCO_3_ Composites Materials

Calcium carbonate composites were synthesized by colloidal crystallization from supersaturated solution via wet-granulation method [39]. CaCl_2_ and Na_2_CO_3_ solution were rapidly mixed in equal volumes. Generally, 0.5 M Na_2_CO_3_ aqueous solution was rapidly poured into an equal volume of 0.5 M solution of CaCl_2_ at room temperature. Then, the precipitate was filtered off by suction filtration after intense agitation on a magnetic stirrer, thoroughly washed with pure water, and dried in vacuum. Chitosan was dissolved in 1% (*w*/*v*) aqueous acetic acid with fast stirring for 12 h to make 1% (*w*/*v*) chitosan solution, which was used as a dispersing phase. Then, the prepared CaCO_3_ powder was dispersed in 200 mL distilled water and poured into 200 mL chitosan solution. The CS–CaCO_3_ composite material were obtained after freeze-drying.

### 2.3. Characterization

The surface morphology of the CaCO_3_ and CS–CaCO_3_ was measured by scanning electron microscopy (SEM, SU8010, Hitachi, Hitachi, Japan). The CaCO_3_ and CS–CaCO_3_ composite material (5 µL) was dropped onto copper grid and dried in the air, then placed on aluminum stub and sputter coated with gold (10 mA for 90 s; JEOL JFC-1600) and imaged at 15 kV after complete air drying.

The X-ray diffraction (XRD) measurements were performed using a diffractometer (XD-3, MSAL, Purk Instrument Co. Ltd., Beijnig, China) with Cu (Ka) radiation at 36 kV and 20 mA. The characteristic XRD peaks of CaCO_3_, chitosan and CS–CaCO_3_ were compared to WH JCPDS (70−2064), International Center for Diffraction Data. (Beijing, China). The Fourier-transform infrared (FTIR) spectra were obtained using an infrared spectrometer (PerkinElmer, Waltham, MA, USA) from KBr pellets at wavelengths ranging from 4000 to 500 cm^−1^ at a resolution of 1 cm^−1^ with an average of 64 scans. The particle sizes were characterized by the DLS (Dynamic light scattering, Zetasizer Ultra, Marvin Co. Ltd., Marvin, UK)

### 2.4. Cell Viability Study

The cell viability of the CaCO_3_ and CS–CaCO_3_ was evaluated by Cell Counting Kit-8 (CCK-8) reduction (Kumamoto City, Japan). Assay with fibroblasts cell line (provided by National Center for Nanoscience and Technology, Beijing, China). Cells were cultured in medium containing 10% fetal bovine serum and 1% penicillin-streptomycin at 37 °C in 5% CO_2_-humidified air. Then 1 mL cell suspension was seeded into a 96-well plate (2 × 10^4^ cells mL^−1^ per well) with medium containing CaCO_3_ and CS–CaCO_3_ with different concentration (0, 0.5, 1, 1.5, and 2 mg mL^−1^). 100 µL CCK-8 was added to each well after incubation for 24 h, and cells were cultured for another 4 h. Absorbance was recorded at 450 nm and cytotoxicity was determined as the percentage of viable cells relative to untreated control cells.

### 2.5. In Vitro Blood Plasma Coagulation Assay

The BALB/C rats (18–20 g) were purchased from the experimental center of the Zhejiang Provincial People’s Hospital, Hangzhou, China. This study was performed in strict accordance with the Guidelines for Care and Use of Laboratory Animals of Zhejiang Provincial People’s Hospital and approved by the Animal Ethics Committee of Zhejiang Provincial People’s Hospital. To explore the activation of CaCO_3_ and CS–CaCO_3_ material-induced blood plasma coagulation, the aPTT (activated partial thromboplastin time), PT (Prothrombin time) and TT (thrombin time)were determined in vitro coagulation assays, which were defined as the activated partial thromboplastin time, prothrombin time and thrombin time, respectively. This experiment was carried out with a full-automatic coagulation analyzer (Werfen Ltd., Birchwood, UK). The venous blood was drawn out from BALB/C rats and mixed with 3.8% (*w*/*v*) sodium citrate with the volume ratio 10:1. The platelet poor plasma (PPP) was obtained by whole blood centrifugation at 3000× *g* rpm/min for 20 min at 4 °C. PT and TT test was performed when 0.5 mL of citrated plasma and 5 mg/mL of simple were mixed and incubated at 37 °C for 5 min and measured by the automatic coagulation analyzer. For aPTT test, the mixture between 0.5 mL of citrated plasma and 5 mg/mL of simple was recalcified with 0.5 mL of 25 mM CaCl_2_ solution.

### 2.6. Blood Coagulation Time

Blood clotting potential of CaCO_3_ and CS–CaCO_3_ composite material systems was examined according to the reference [26]. 100 mg of CaCO_3_ and CS–CaCO_3_ composite material were put into the 48-well plate. Then 50 µL of 25 mM CaCl_2_ solution was added to 250 µL calcified blood, 300 µL of the rat blood solution prepared was added into the wells. The wells were then washed using 0.9% NaCl solution to remove the unclotted blood. The blood clotting time was defined as the time at which the formation was uniform and there were stable clots in the wells.

### 2.7. Adherence of RBC (Red blood cell) and Platelet on CaCO_3_ and CS–CaCO_3_ Composite Material

The BALB/C rat blood with citrate was added with 1 mg of CaCO_3_ or the CS–CaCO_3_ composite material and then incubated at 37 °C for 30 min. The precipitates were filtered and washed 3 times with PBS (phosphate buffered saline), and then 2.5% glutaraldehyde solution was added to the precipitates and kept at room temperature. After 2.5 h, the precipitates were washed with PBS and dehydrated in ethanol solution with different concentration (20%, 40%, 60%, 80% and 100%), and then dried in the air. The BALB/C rat blood was added with citrate and then the Platelet Rich Plasma (PRP) was seperated by centrifuging at 2000 rpm/min for 20 min. The upper PRP layer was taken out by pipette and added into the cuvet containing CaCO_3_ or CS–CaCO_3_, and then they were incubated at 37 °C for 30 min. The precipitation was collected and treated in the same way with RBCs. The adhesion and morphology of the platelet CaCO_3_ and CS–CaCO_3_ composite material were observed by SEM after being coated with gold.

### 2.8. Water Absorption Efficiency In Vitro

CaCO_3_ and CS–CaCO_3_ composite materials were weighed as the same mass and immersed in the PBS buffer with pH 7.4 at 37 °C overnight. The CaCO_3_ and CS–CaCO_3_ composite materials were taken out from PBS buffer at same time intervals, then the water was wiped up with filter paper gently and immediately. The water absorption ratio (A) was calculated by:$$A=\frac{M_{W}-M_{d}}{M_{d}}\times 100\%$$

In the above equation, *A* is swelling ratio, *M_w_* is the wet weight of the materials and *M_d_* is the dry weight of the materials.

### 2.9. Hemolysis Assay

The BALB/C rat blood was transferred into 3.8% sodium citrated solution. The collected blood was centrifuged at 2000× *g* rpm/min to separate the RBCs. Then one milliliter of concentrated RBCs was diluted with saline solution. The hemolytic activity of the CaCO_3_ and CS–CaCO_3_ with different concentrations (0.5, 1.0, 2.0, 3.0, and 5.0 mg/mL) were examined. The samples in 0.1% Triton-X (100 µL) were used as positive control and the samples in saline (100 µL) were used as negative control. Diluted 2% *w*/*v* RBCs (500 µL) was added and incubated at 37 °C for 1 h. Then the samples were centrifuged (3500× *g* rpm; 10 min) to obtain the supernatant. The collected supernatant was determined by spectrophotometry as having an absorbance value of 540 nm using a well plate reader. Percent hemolysis (%) was calculated following the equation as below:$$\mathrm{Hemolysis}\;\%=\frac{A_{\mathrm{sample}}-A_{\mathrm{PBS}}}{A_{\mathrm{water}}-A_{\mathrm{PBS}}}\times 100\%$$

In this expression, A_sample_, A_PBS_, A_water_ represent the absorbance value of test samples, positive (water), and negative (PBS) controls

### 2.10. In Vivo Wound Healing

The rats used to assess hemostatic performance were divided into five groups, each group contains three rats. Every group of rats took normal feed and water without restriction. The rats were anesthetized by intramuscular injection with 1% pentobarbital sodium. The dorsal area of the rats was depilated and cleaned with alcohol. A wound of 1 cm^2^ was produced by excising the skin of the rat. The rats with bare wounds were used as a negative control, and the other rats with the wounds were covered with CaCO_3_ and CS–CaCO_3_ to measure the wound healing property of the novel material. After covering the materials, the rats were returned individually in cages under normal room temperature. The covering materials were changed at day 1, 2, and 3, and the wound area was measured with a ruler and photographs were taken. The rats were recovered in 2 weeks to be used for evaluating wound-healing properties of the CaCO_3_, and CS–CaCO_3_. 

The percent reduction in wound size was calculated as the formula as below:$$\mathrm{reduction}\%=\frac{W_{0}-W_{t}}{W_{0}}\times 100\%$$

In this expression, W_0_ and W_t_ represent the wounds size on the 1st day and the 7th day, respectively.

After day 14, the skin wound tissue of the rat was excised and fixed with 10% formalin to be stained with a hematoxylin−eosin. Histological analysis was measured by H&E and Masson’s trichrome.

## 3. Result and Discussion

### 3.1. Characterization of the Prepared of CaCO_3_ and CS–CaCO_3_ Composites Materials

CaCO_3_ and CS–CaCO_3_ were synthesized by colloidal crystallization from supersaturated solution and obtained after freeze-drying. SEM images of the prepared CaCO_3_ showed spherical-shaped morphology and with particle size of 3 µm (Figure 1a). In Figure 1b,c, the CS–CaCO_3_ displays spherical-shaped particles and an increased surface roughness, respectively, which was attributed to surface complexation resulting from the complexation of CaCO_3_ and chitosan. In Figure 2, the DLS (Dynamic light scattering) results exhibit the particle radius of CaCO_3_ and CS–CaCO_3_ of about 600 nm, and the polydispersity index of CaCO_3_ and CS–CaCO_3_ are 1.34 and 1.22, respectively; these results are in accordance with the SEM results and indicate remarkable monodispersion of the synthesized CaCO_3_ and CS–CaCO_3_.

Presence of CaCO_3_ and CS–CaCO_3_ in the novel material was further illustrated by FTIR. FTIR data of the developed CS–CaCO_3_ composite material (Figure 3a) showed the characteristics peaks of CaCO_3_ at 713, 877, and 1427 cm^−1^ (−CO_3_ stretching) and characteristics peaks of Chitosan at 1070 (−C−O stretch), 1425, 1566 (−N−H stretch) and 2920 cm^−1^ (−C−H stretch), which indicated the presence of CaCO_3_ in the CS–CaCO_3_ composite materials.

The CaCO_3_, chitosan, and CS–CaCO_3_ were further characterized by XRD analysis. The XRD peaks obtained for the CaCO_3_ particles matched with the XRD peaks of CaCO_3_ that corresponded to calcite (The Inorganic Crystal Structure Database (ICSD) no. 158257) [28]. It is evident that native Chitosan had characteristic broad peaks at 21.08, which are typical of a chitosan structure. In the XRD spectra of CS–CaCO3, the characteristic peak of CaCO_3_ at 2θ of 23.44°, 29.76°, 36.34°, 39.8°, 43.5°, 47.84°, 48.94° was observed (Figure 3b). This further confirmed the presence of CaCO_3_ in the composite materials. The crystallinity of CaCO_3_ was higher than CS–CaCO3. There was almost no difference in crystallinity between CaCO_3_ and CS–CaCO_3_. The result indicated that complexation did not disrupt the ordered structure of the crystalline regions in CaCO_3_. The CS–CaCO_3_ was successfully fabricated by complexation between amino groups and Ca^2+^ [40].

### 3.2. Fluid Sorption Capability

CaCO_3_ and CS–CaCO_3_ were studied for water absorption for 24 h at room temperature. Figure 4 shows the water absorption ratio of CaCO_3_ was 124%, however, the water absorption ratio for chitosan and CS–CaCO_3_ can reach 345% and 459%, which is much higher than that of CaCO_3_. The water absorption rate of CaCO_3_ was low, which can be attributed to only a small surface area of inorganic compound CaCO_3_. The water absorption amount increased with the porous structure formation on the material surface. The water absorption of pure can get to 350 % because of the capillary action. However, the equilibrium water absorption amount of CS–CaCO_3_ was the highest, which may be contributed to by the CaCO_3_ particles having high surface area and the combination of the chitosan capillary action.

### 3.3. Cytotoxicity Tests of the CS–CaCO_3_

A good hemostatic agent must have high biocompatibility and low cytotoxicity. Therefore, fibroblast cells were treated with various concentrations of CaCO_3_ and CS–CaCO_3_ to measure the cytotoxicity of the novel hemostatic material. Cell viability was achieved using CCK-8 assay and live/dead viability assay. Figure 5 shows that both CaCO_3_ and CS–CaCO_3_ could significantly promote cell viability, which demonstrates that CaCO_3_ and CS–CaCO_3_ have high biocompatibility and low cytotoxicity and have the potential application for hemostatic materials.

### 3.4. Blood Clotting Activity 

Blood clotting time measurement can be used to reflect the hemostatic potential of a novel material. As shown in Figure 6a, compared with control and CaCO_3_, blood clotting time decreased greatly for the sample added with the CS–CaCO_3_. The results of blood clotting time indicated that CS–CaCO_3_ might have better hemostatic performance. Hemostasis can prevent a large amount of blood loss after vascular injury. It depends on a complex process including platelets, other cells and the activation of specific blood proteins known as coagulation factors.

In vitro activated partial thromboplastin time (aPTT), thrombin time (TT) and prothrombin time (PT) were measured to estimate the hemostasis. Figure 6 shows the APPTs, TTs and PTs of samples results. Compared with the control, the aPTT of CS–CaCO_3_ at the concentration of 0.5 mg/mL and 1.0 mg/mL kept the same but the aPTT of CaCO_3_ dropped from 54 to 50 s. For both of the concentrations, the aPTT of CS–CaCO_3_ is shorter than CaCO_3_ (Figure 6b). Figure 6c,d shows that the TTs and PTs of samples follow a similar trend, and no significant dosage-dependent relationship was observed for both CaCO_3_ and CS–CaCO_3_. The results illustrate that the CS–CaCO_3_ could not only decrease the time of thromboplastin formation, but also accelerate the activation of the intrinsic coagulation system. The addition of CaCO_3_ and CS–CaCO_3_ accelerate the coagulation system.

### 3.5. Hemolytic Activity

A hemolysis assay was used to determine the toxicity of CS–CaCO_3_ on RBCs. A total of 2.0 mL of isolated RBCs was mixed with CaCO_3_ and CS–CaCO_3_ with different concentrations in PBS at a range of 0.5 to 2.0 mg mL^−1^ and incubated at room temperature for 1 h. Additionally, positive and negative control samples were prepared by adding 2.0 mL of water and PBS respectively to RBCs. At every 30 min, precipitated cells and particles were resuspended by gentle shaking. The photographs showing precipitated RBCs at the end of the hemolysis experiment are given in Figure 7a. The red color of the released hemoglobin from damaged cells is clearly observable for the positive control. For CaCO_3_ and CS–CaCO_3_, the supernatants are almost colorless at all concentrations, which illustrates the CaCO_3_ and CS–CaCO_3_ can almost completely prevent the hemolytic activity.

Hemolytic activity of the CS–CaCO_3_ was determined by measuring the absorption peak of hemoglobin at 540 nm, which was released to the solution from the hemolyzed cells. Hemolysis results for both CaCO_3_ and CS–CaCO_3_ are given in Figure 7. CaCO_3_ showed the higher hemolytic activity than CS–CaCO_3_, which was observed for all the experiment concentrations. On the other hand, the hemolytic activity of CaCO_3_ kept the value between 1.4~1.9%; however, the hemolytic activity of CS–CaCO_3_ increased with the concentration. The photographs showing precipitated RBCs at the end of the hemolysis experiment are given in Figure 7a. The red color of the released hemoglobin from damaged cells is clearly observable for CS–CaCO_3_. For CS–CaCO_3_ the supernatants are almost colorless at all concentrations. Therefore, the percentage of hemolysis of the CaCO_3_ and CS–CaCO_3_ was found to be less than 5%, which is an allowable limit for biomaterials, which exhibited that CaCO_3_ and CS–CaCO_3_ composite materials were hemocompatible.

### 3.6. Platelet Adhesion

The platelet adhesion property of the CaCO_3_ and CS–CaCO_3_ was determined by agitation of the samples, and then SEM was conducted to observe the capture of adherent platelets of prepared samples (Figure 8). The SEM images show the platelets adhered to the CS–CaCO_3_ and illustrate the morphology of the adhered platelets being exposed to the materials after 30 min. The surfaces of CaCO_3_ and CS–CaCO_3_ were both covered with a lot of adhered platelets. It can be estimated that the platelet aggregation was stimulated by CaCO_3_ and CS–CaCO_3_. The result illustrates that CaCO_3_ and CS–CaCO_3_ promoted the aggregation and activation of platelets. At the same time, RBCs were observed with good dispersity and biconcave shape. With the addition of CaCO_3_ and CS–CaCO_3_, RBCs joined together with roughly linear chains, which demonstrates CaCO_3_ and CS–CaCO_3_ can cause RBC aggregation.

### 3.7. Wound Healing

Last, the prepared CaCO_3_ and CS–CaCO_3_ was conducted in vivo to prove the enhanced wound healing ability. We examined the wound-healing properties of CaCO_3_ and CS–CaCO_3_ in vivo by treating a small cut (1 cm × 1 cm) with CaCO_3_ or CS–CaCO_3_ for 3, 5, 7, and 9 d, and bare wounds were used as the control (Figure 9). The extent of wound healing was evaluated macroscopically. Wounds were almost completely healed after 10 d which were treated with CS–CaCO_3_. Compared to the control group, the CaCO_3_ and CS–CaCO_3_ treatment group both exhibited a more rapid recovering of wound size, especially in the first 7 d (Figure 9a).

Furthermore, qualitative histomorphologies stained by hematoxylin and eosin sections were studied. The wound healing was comprised of three stages: inflammation, new tissue formation, and remodeling healing. Histological analysis showed the extent of collagen deposition of the wound treated with CaCO_3_ and CS–CaCO_3_, which was confirmed by H&E-stained images (Figure 10). New blood vessels and glandular cavity was shown in wounds treated with CaCO_3_ and CS–CaCO_3_, which illustrated cell proliferation and new tissue formation in the wound. The epithelial tissue of the CS–CaCO_3_ wound after 10 d revealed the best wound healing properties compared to those treated with CaCO_3_ and without treatment. The collagen content of the wound area was studied by Masson’s trichrome staining. The density of the collagen fiber in the wounds was increased upon treatment with CaCO_3_ and CS–CaCO_3_ as compared to the control group (Figure 10). These results indicate that CS–CaCO_3_ are not only procoagulant but also have wound healing capacity.

## 4. Conclusions

In conclusion, we developed a CS–CaCO_3_ composites material which can be used for hemorrhage control and wound healing. Compared with single CaCO_3_, the CS–CaCO_3_ composite material showed shorter bleeding time and better wound-healing performance. Otherwise, the CS–CaCO_3_ composite material also exhibited a stronger fluid absorbance property of concentrated RBCs, blood platelets, and clotting factors. The CS–CaCO_3_ composite material activated the intrinsic coagulation cascade, showing better hemorrhage control, biocompatibility and wound healing property than the CaCO_3_ granules. Our results illustrated that the CS–CaCO_3_ composite material can be used to prevent massive blood loss and that it is an effective hemostatic agent.

## Data Availability

The data reported in this article are available on request from the corresponding author.

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
