# Peer review of "CaCO3–Chitosan Composites Granules for Instant Hemostasis and Wound Healing"

_materials, 2021, doi:10.3390/ma14123350_

Round 1
Reviewer 1 Report
The manuscript entitled „ Novel CaCO3-Chitosan Composites Granules for Instant He-2 mostasis and Wound Healing” authored by Wei He, Xiaodong Huang, Jun Zhang, Yue Zhu, Po Li, Yajun Liu, Bo liu, Qilong Wang, Xiaonan Huang and Da He comprises a novel developed CaCO3 and chitosan composite material used for hemorrhage control and wound healing:
In my opinion, the manuscript requires major changes/additions:
- First of all, I am not a native English speaker, but I think that due to the English formulations, the article is really hard to read and understand. I suggest a major revision for English writing.
As an example, in the abstract section, the authors said: “The rat injury model being therapy by CS-CaCO3 composites demonstrated aid in rapid wound healing.” I don’t understand.
Another example, in the Introduction: “….lot of hemostatic material are used for hemorrhage controlling [3,4], which can be divided into inorganic 31 materials [5–7]and polymer materials [8].” From what I now, polymers can be either organic or inorganic or a combination. So I don’t understand what the authors want to say.
Again, in the Introduction: “However, the main disadvantage for applying the chitosan material for severe trauma is that chitosan is not sufficiently flexible to fill large or deep wounds. Otherwise, the appearance of chitosan granules may induce the risk of thromboembolic complication in vessels [36].”, I don’t understand…
And I many other examples…
Overall, the manuscript is clearly structured and comprises adequate analyses.
Author Response
1.In the abstract section, the authors said: “The rat injury model being therapy by CS-CaCO3 composites demonstrated aid in rapid wound healing.” I don’t understand.
Line 23 : The language presentation was improved in the manuscript with the highlight as “According to the results of wound healing experiments in rats, it can be found that CS-CaCO3 composite material can promote wound healing.”
2.In the Introduction: “….lot of hemostatic material are used for hemorrhage controlling [3,4], which can be divided into inorganic 31 materials [5–7]and polymer materials [8].” From what I know, polymers can be either organic or inorganic or a combination. So I don’t understand what the authors want to say.
Line 32 : The revised statement is as follows“A lot of hemostatic material are used for hemorrhage controlling , which can be divided into inorganic composite material and organic polymer composite material.”
3.Again, in the Introduction: “However, the main disadvantage for applying the chitosan material for severe trauma is that chitosan is not sufficiently flexible to fill large or deep wounds. Otherwise, the appearance of chitosan granules may induce the risk of thromboembolic complication in vessels [36]”, I don’t understand
Line63:We have added the revised content to the manuscript as” However, chitosan hemostatic materials cannot be used to treat wounds with heavy bleeding, because chitosan materials have no adhesion and may be washed away by blood, or chitosan materials may enter the blood vessels of the wound and cause thrombosis.”

Reviewer 2 Report
The article, entitled "Novel CaCO3-Chitosan Composites Granules for Instant Hemostasis and Wound Healing" by authors He et al, describes the use of a chitosan-based compound in wound therapy. The results obtained are promising and acceleration of wound healing is an important factor in the healing and recovery process.
Several improvements and clarifications are needed for the paper to be published.
The authors should not use the term "novel" in relation to CaCO3-Chitosan Composites. It has been previously described as a material used in stopping bleeding, e.g. Polymer 123 (2017) 194-202; Mar Drugs. 2018 Aug; 16(8): 273; Polym Adv Technol. 2019; 30: 143– 152 and many others (please check older ones ). Yes, the results of the animal studies are original but the substance used is no longer.
The authors should present and describe the chemical structure of CaCO3-Chitosan Composites. How is calcium incorporated into chitosan structures? Does the chitosan undergo crosslinking? A schematic drawing would be useful here. The images shown in Figure 1 do not provide an answer here.
Line 148: the abbreviation PBS is not developed until line 160.
Lines 164 and 165: inconsistent designations of physical quantities (mass: M or W).
Line 179: the formula should be rewritten in non-linear form, as in line 164.
Line 183-184: "The mice used to assess hemostatic performance were divided into five groups, each group containing three rats." Please decide: Mice or rats?
Line 194: What are: W_0 and W_t?
Line 207: "DLS results exhibited the particle size distributions of ..." Please show the DLS results (is DLS Dynamic Light Scattering?)
Lines 232-234: Since pure CaCO3 does not contain chitosan, how does the surface charge depend on the acetic acid group on chitosan?
Figure 4: is the water absorption value of CS-CaCO3 more or less the sum of the absorption values of chitosan and CaCO3 separately? This would mean that the chitosan and CaCO3 is a simple mixture and not a composite with chemical bonds between the components.
Line 271: Is "Figure 6 shows the PTs and aPTTs of sample results", shouldn't it be: "Figure 6 shows the PTs, TTs and aPTTs of samples results" as in the caption to Figure 6?
Lines 286-287: "with different concentrations in PBS at a range of 0.5 to 5 mg/mL". Figure 7b shows concentrations from 0.5 to 2.0 mg/mL.
Lines 290 and 301: repetition?
Line 301: "... concentdration. the ..." -> "... concentration. The ..."
Line 317: "The result illustrates CS-CaCO3 promoted the aggregation and activation of platelets." It can't see this effect in Figure 8. If both pictures were at the same scale maybe it would see the differences. Similarly, the scale problem can be seen in Figure 1.
Linia 337: "H&E(up) and Masson’s trichrome(down)." -> "H&E(left) and Masson’s trichrome(right)."
Linie 21 oraz 353: font size is inappropriate (subscript?)
Author Response
Thanks to the reviewer’s kind suggestion for the modification suggestion, and we have made relative correction of the manuscript.

Reviewer 3 Report
This manuscript is focused on the preparation and characterization of CaCO3-chitosan composite as a potential hemostatic agent. The paper could have some scientific contribution, but it has to be improved. The material part must be described and presented in a reliable and accurate manner allowing others to repeat the presented research.
Specific comments:
- Section 1. Introduction: The authors do not provide sufficient background on using chitosan-based materials as hemostatic agents. Although chitosan-based hemostatic dressings (e.g., HemCon, Axiostat® Military, and some others) have been approved by the US Food and Drug Administration for hemorrhage control and have successfully been used for both civilian and military trauma patients. I recommend that the authors add the analysis of chitosan-based hemostatics to the Introduction.
- Line 52: Replace “natural polymer” with “polymer of natural origin” or “natural-origin polymer” (Strictly speaking, chitosan is not natural, but semi-synthetic polymer).
- Section 2.1. Chemicals: The chitosan sample must be carefully characterized regarding its molecular weight (by viscometry, light scattering, or size exclusion chromatography) and the degree of deacetylation (by NMR, IR, elemental analysis, or titration). The properties of chitosan are very dependent on these parameters; therefore, the wide ranges of values provided by the manufacturer (like Mw: 100,000–300,000 Da; line 80) are clearly insufficient. Also, indicate the source of chitosan (crab, shrimp, fungi, etc.). The European Chitin Society does not recommend the publication of the research papers on chitosan, in which this information is missing.
- Line 232-234: This statement is incorrect. First, the zeta potential of the original CaCO3 cannot be determined by the properties of chitosan. Second, the zeta potential of chitosan in acetic acid solution is positive due to the protonation of the amino group. Please analyze how the zeta potential of the CaCO3-CS composite changes compared to the original CaCO3.
- The quality of Figures 2 and 3 must be improved; they are hardly visible.
- Figure 2a: The Transmittance is usually expressed in %. What is the unit of measurement in this case (given a range of 0.4-1.3)?
- Line 19: Round off 458.8% to the nearest whole number.
- Line 209: Add here a reference to Figure 3b and provide the average hydrodynamic diameters and polydispersity indexes for both CaCO3 and CaCO3-CS.
- The English writing style could have been better. There are too many English errors and wrong constructions, which sometimes drastically affect the scientific meaning, making the paper difficult to read and understand. A native English speaker with a scientific background should carefully revise the manuscript before its resubmission.
- Check the appropriateness of using acronyms. Overuse or inappropriate use of acronyms makes reading paper difficult.
- Do not introduce an acronym unless you will use it. To keep using acronyms to a minimum, only insert an acronym if the term is used at least three times (e.g., DLS, line 207).
- Once you introduce the acronym, use it consistently after that (e.g., you used chitosan 45 times in the text, even though you introduced the acronym CS).
Author Response
Thanks to the reviewer's kind suggestion and we have modified the manuscripts.

Round 2
Reviewer 1 Report
As the authors adressed the suggestions, I recommend the acceptance of the manuscript in present form
Author Response
Thank s for the reviewer.
Reviewer 2 Report
Not all technical issues (font size, repetitions) have been corrected. Careful technical proofreading needed.
Author Response
The manuscript has been modified.
Reviewer 3 Report
To my regret, the authors did not address most of my comments, so I have to repeat most of them. Also, the authors did not provide answers to my questions and comments (only the manuscript file is attached). I am pretty disappointed in this revision of the paper.
- Section 2.1. Chemicals: The chitosan sample must be carefully characterized regarding its molecular weight (by viscometry, light scattering, or size exclusion chromatography) and the degree of deacetylation (by NMR, IR, elemental analysis, or titration). The properties of chitosan are very dependent on these parameters; therefore, the wide ranges of values provided by the manufacturer (like Mw: 100,000–300,000 Da or degree of deacetylation >75%, lines 81-82) are clearly insufficient. The European Chitin Society does not recommend the publication of the research papers on chitosan, in which this information is missing.
- Line 243-244: The zeta potential of chitosan in acetic acid solution (or chitosan acetate) is positive due to the protonation of the amino group and cannot be “caused by the acetic acid groups on Chitosan”. Please analyze how the zeta potential of the CaCO3-CS composite changes compared to the original CaCO3 and chitosan acetate.
- Figure 3a: The Transmittance is usually expressed in %. What is the unit of measurement in this case (given a range of 0.4-1.3)?
- Line 19: Round off 458.8% to the nearest whole number as in line 250.
- Line 219: Provide the average hydrodynamic diameters and polydispersity indexes for both CaCO3 and CaCO3-CS.
- The English writing style could have been better. There are too many English errors and wrong constructions, which sometimes drastically affect the scientific meaning, making the paper difficult to read and understand. A native English speaker with a scientific background should carefully revise the manuscript before its resubmission.
- Scheme 1 must be reconsidered. Calcium ions do not tend to form coordination bonds with amino groups.
- Lines 238-239: “The CS-CaCO3 was successfully fabricated by utilizing the strong chelation between carboxyl groups and Ca2+.” The authors give a contradictory interpretation by showing calcium coordination of amino groups in Scheme 1 and describing chelation with carboxyl groups in the text. I also want to note no strong chelation between calcium ions and carboxyl groups (of acetic acid?). Check also the use of the term chelation (https://en.wikipedia.org/wiki/Chelation).
- Check the numbering of the figures. There are two Figures 4 in the text (P.7), incorrect reference to Figure 3a (line 243).
Author Response
The response has been added in the word.

Round 3
Reviewer 3 Report
I still cannot recommend the paper for publication because a few major problems have not been addressed in the revised manuscript:
- The authors specify very large ranges of chitosan characteristics (Mw: 100,000–300,000 Da, from shrimp shells, deacetylated ratio ≥75%). I would like to point out that chitosan samples with a molecular weight of 100 and 300 kDa and a degree of deacetylation of 75 and 100% give a huge difference in properties. Therefore, I cannot accept the authors' excuse based on examples from the literature. If you work with natural polymers, you must be prepared to characterize each sample by its essential parameters.
- The scale in Figure 3 is in the range 0.4-1.3 and cannot be in % (this range is within the baseline). Please refer to the original data and adjust the scale.
- Remove zeta-potential measurements (lines 109-111).
- The authors use the term "chelation" incorrectly. Complexation of calcium with chitosan (Scheme 1) do not result in the formation of any chelate rings.
Author Response
- The authors specify very large ranges of chitosan characteristics (Mw: 100,000–300,000 Da, from shrimp shells, deacetylated ratio ≥75%). I would like to point out that chitosan samples with a molecular weight of 100 and 300 kDa and a degree of deacetylation of 75 and 100% give a huge difference in properties. Therefore, I cannot accept the authors' excuse based on examples from the literature. If you work with natural polymers, you must be prepared to characterize each sample by its essential parameters.
The relative characterization was added in the part of material with highlight.
2.The scale in Figure 3 is in the range 0.4-1.3 and cannot be in % (this range is within the baseline). Please refer to the original data and adjust the scale.
The scale has been improved.
3.Remove zeta-potential measurements (lines 109-111).
The data has been deleted.
4.The authors use the term "chelation" incorrectly. Complexation of calcium with chitosan (Scheme 1) do not result in the formation of any chelate rings.
The chelating has been changed to complexation.
